# Effectiveness of the second COVID-19 booster against Omicron: a large-scale cohort study in Chile

Alejandro Jara[1,2], Cristobal Cuadrado[3,4], Eduardo A. Undurraga ®[5,6,7,8], Christian García[3], Manuel Nájera[3], María Paz Bertoglia ®[3], Verónica Vergara[3], Jorge Fernández ®[3], Heriberto García-Escorza[3] & Rafael Araos ®[6,9,10] ✉

In light of the ongoing COVID-19 pandemic and the emergence of new SARS-CoV-2 variants, understanding the effectiveness of various booster vaccination regimens is pivotal. In Chile, using a prospective national cohort of 3.75 million individuals aged 20 or older, we evaluate the effectiveness against COVID-19-related intensive care unit (ICU) admissions and death of mRNA-based second vaccine boosters for four different three-dose background regimes: BNT162b2 primary series followed by a homologous booster, and CoronaVac primary series followed by an mRNA booster, a homologous booster, and a ChAdOx-1 booster. We estimate the vaccine effectiveness weekly from February 14 to August 15, 2022, by determining hazard ratios of immunization over non-vaccination, accounting for relevant confounders. The overall adjusted effectiveness of a second mRNA booster shot is 88.2% (95%CI, 86.2–89.9) against ICU admissions and 90.5% (95%CI 89.4–91.4) against death. Vaccine effectiveness shows a mild decrease for all regimens and outcomes, probably linked to the introduction of BA.4 and BA.5 Omicron sub-lineages and the waning of immunity. Based on our findings, individuals might not need additional boosters for at least 6 months after receiving a second mRNA booster shot in this setting.

The COVID-19 pandemic has caused more than 615 million cases and 6.5 million deaths reported globally as of September 2022[1]. COVID-19 vaccines have been essential to decrease the burden of disease and reduce restrictions associated with the pandemic response. A robust body of evidence showed that the primary series of several COVID-19 vaccines had high efficacy and effectiveness against symptomatic COVID-19 and severe illness in the first months[2,3]. However, emerging evidence of the additional protection of a booster dose, the

emergence of new, highly transmissible SARS-CoV-2 lineages with increased immune evasion, and suggestive evidence of immunity waning led many countries to roll out booster shots 4–6 months following the primary series[4–7]. Research showed that recently administered homologous or heterologous COVID-19 boosters restored waning protection against symptomatic infection and severe illness[8–10].

The emergence and spread of the B.1.1.529 Omicron variant of SARS-CoV-2, and its sub-lineages, caused an unprecedented global

[1]Facultad de Matemáticas, Pontificia Universidad Católica de Chile, Santiago, Chile. [2]Center for the Discovery of Structures in Complex Data (MiDaS), Santiago, Chile. [3]Ministerio de Salud de Chile, Santiago, Chile. [4]School of Public Health, Universidad de Chile, Santiago, Chile. [5]Escuela de Gobierno, Pontificia Universidad Católica de Chile, Santiago, RM, Chile. [6]Multidisciplinary Initiative for Collaborative Research in Bacterial Resistance (MICROB-R), Santiago, Chile. [7]Research Center for Integrated Disaster Risk Management (CIGIDEN), Santiago, Chile. [8]CIFAR Azrieli Global Scholars program, CIFAR, Toronto, Canada. [9]Instituto de Ciencias e Innovación en Medicina, Facultad de Medicina, Universidad del Desarrollo, Santiago, Chile. [10]Advanced Center for Chronic Diseases (ACCDiS), Santiago, Chile. ✉e-mail: rafaelaraos@udd.cl

surge in COVID-19 cases[1,11]. This rise may be partly explained by Omicron's high transmissibility and ability to partially evade natural and vaccine-induced immunity against SARS-CoV-2[12,13] and from the waning of vaccine protection[14–16]. While booster doses restored a substantial amount of protection[8,9,17,18], further research showed substantial reductions in the protection of boosters against symptomatic COVID-19[19–21], as happened with the primary series of COVID-19 vaccines. Early in 2022, several countries began rolling out second boosters (a fourth dose for most vaccines) 4–6 months following the first booster dose, and international organizations have recommended its use for at-risk populations[22–26]. Policymakers need evidence of real-world effectiveness to guide vaccination policies. However, there is limited evidence for the effectiveness of second boosters, and it primarily refers to mRNA vaccines in Canada, Israel, Korea, Portugal, Sweden, and the USA among populations at higher risk (e.g., older adults and immunocompromised persons)[21,24,25,27–32]. Furthermore, a few studies have examined the potential waning of protection against COVID-19 for second boosters[27] or the effectiveness against severe disease of a second booster dose for individuals who received their primary series based on inactivated vaccines[26]. A study in Thailand found no severe outcomes, including death, in patients who received a fourth COVID-19 vaccine dose among adults with heterologous three-dose vaccine regimes, including inactivated virus vaccines[26]. A considerable proportion of individuals in low- and middle-income countries received their primary series based on inactivated vaccines[33].

Using a large prospective national observational cohort in Chile, we evaluated the effectiveness of mRNA-based second vaccine boosters for individuals with four different three-dose background regimes: (1) BNT162b2 primary series plus a homologous booster (3mRNA), (2) CoronaVac primary series plus mRNA booster (CC+mRNA), (3) CoronaVac primary series plus homologous booster (CCC), and (4) CoronaVac primary series plus ChAdOx-1 booster (CCA). We estimated vaccine effectiveness weekly, from February 14, 2022, to August 15, 2022, against admission to an intensive care unit (ICU) and death (U07.1) associated with laboratory-confirmed SARS-CoV-2 infection. We estimated the overall vaccine effectiveness of an mRNA second booster, regardless of the three-dose background regimen, and for each group, using survival regression models to estimate hazard ratios of immunization (>13 days after the second dose) over non-vaccination and compared to three-dose background regimes, accounting for time-varying vaccination exposure and clinical, demographic, and socioeconomic confounders at baseline.

The Chilean Ministry of Health launched the second booster campaign on February 14, 2022, based on a standard dose of Pfizer-BioNTech's BNT162b2 or Moderna's mRNA-1273 vaccines. All adults were eligible for a second booster dose, although priority was given to older people, front-line health workers, immunocompromised individuals, and persons with underlying conditions associated with the risk of severe COVID-19. By August 29, 2022, more than 10.6 million individuals had received a second booster dose, representing 71.7% of the target population[34]. COVID-19 is a notifiable disease in Chile; all suspected cases are notified to health authorities through Epivigila, an electronic surveillance system, and undergo laboratory confirmation. RT-PCR and antigen tests are freely available for FONASA affiliates in healthcare centers throughout the country in a network of primary care centers and referral hospitals. FONASA does not discriminate by age, sex, income, number of dependents, pre-existing conditions, or nationality. The government tracks vaccination schedules through an electronic national immunization registry, and vaccination rollout was organized using a publicly available national vaccination schedule that assigns specific vaccination dates to eligible groups. Eligible individuals show up at their nearest vaccination site (e.g., primary healthcare clinic) with an ID; they do not need an appointment. A minimum of 20 weeks were required between the first and the second booster dose. Many non-pharmaceutical interventions to control COVID-19 (e.g.,

school closures) were no longer enforced during the study period, although health authorities still recommended washing hands frequently, adequate ventilation, and physical distancing in public places. Confirmed COVID-19 cases were required to stay at home and face-masks were mandatory in public transport and other public spaces throughout the study period. We provide additional descriptions of the COVID-19 vaccination campaigns and the Chilean healthcare system elsewhere[8,35].

Our study cohort included adults aged ≥20 years and affiliated with the Fondo Nacional de Salud (FONASA), the public national healthcare system, who completed a CoronaVac or BNT162b2's two-dose primary series at least 120 days before the beginning of the follow-up on August 11, 2021, when the first booster campaign was launched, and unvaccinated individuals. The study cohort included immunocompromised individuals. We excluded individuals with confirmed COVID-19 according to reverse-transcription polymerase-chain-reaction assay for SARS-CoV-2 or antigen test reported before August 11, 2021 (Supplementary Fig. S1). We excluded individuals who received three doses of CoronaVac plus a homologous CoronaVac second booster due to the small sample size ($n = 1788$). We also excluded individuals with a second vaccine booster who had received a primary series of Oxford-AstraZeneca's ChAdOx1 adenoviral vector vaccine or CanSinoBIO's Ad5-nCoV ($n = 12,241$).

## Results

Our final cohort included 3,754,785 adults. Of these, 2,623,802 (69.9) received a second booster shot of BNT162b2 or mRNA-1273 vaccine between February 14, 2022, and August 15, 2022, and 757,726 (20.2%) had not been vaccinated by the end of the follow-up. Vaccination rollout was organized through a publicly available schedule and was free of charge (Supplementary Fig. S2). Notably, there was a high level of SARS-CoV-2 circulation during the rollout, in which the predominant Omicron sub-lineages were BA.2.1.12, BA.4, and BA.5 (Supplementary Fig. S3). Cohort characteristics are described in Supplementary Tables S1–S5. The incidence of COVID-19 and the vaccination status at the end of the follow-up differed significantly ($P < 0.001$) by participant's sex, age, comorbidities, nationality, region of residence, and income.

The overall age- and sex-standardized incidence of COVID-19-related ICU admissions and deaths among unvaccinated individuals was 0.29 per 100,000 person-days (95% confidence interval CI 0.26–0.32) and 0.68 (95% 0.64–0.75), respectively. In contrast, the adjusted incidence for ICU admissions and deaths for participants with a second booster dose was 0.05 per 100,000 person-days (95% CI 0.04–0.07) and 0.08 per 100,000 person-days (95% CI 0.07–0.09) (Table 1).

At the end of follow-up, the overall adjusted vaccine effectiveness of a second booster to prevent ICU admission and death was 88.2% (95% CI 86.2–89.9) and 90.5% (95% CI 89.4–91.4), respectively. These estimates represent a moderate but significant reduction in vaccine effectiveness compared to the maximum observed in the cohort during the study period of 96.8% (95% CI 86.8–99.2) for the prevention of ICU admission and 96.7% (95% CI 93.0–98.3) for the prevention of COVID-19-related death (Fig. 1 and Supplementary Tables S6 and S7). To understand additional protection conferred by the second booster compared to the first booster, we provide overall estimates of vaccine effectiveness against COVID-19-related ICU admission and death for adults >20 years of mRNA-based second vaccine boosters compared to three-dose background regimes (Supplementary Table S8 and Supplementary Fig. S4). The results show that a second vaccine booster effectively prevents COVID-19-related ICU admission and death and provides additional protection compared to the first booster.

The scheme-specific adjusted vaccine effectiveness of a second booster dose against COVID-19-related ICU admission at the end of follow-up was 74.2% (95% CI 54.8–85.2), 77.0% (95% CI 69.7–83.0), 75.0%

**Table 1 | Effectiveness against ICU admissions and death of mRNA-based second vaccine boosters in adults aged 20 years and older, February 14, 2022, through August 15, 2022***

| Immunization status | Persons at risk | Person-days | Cases | | Vaccine effectiveness (95% CI) |
|---|---|---|---|---|---|
| | | | No. | Age and sex-adjusted incidence (100 thousand person-days) | End of follow-up (August 15, 2022) |
| **Admitted to ICU** | | | | | |
| Unvaccinated | 749,856 | 137,442,303 | 388 | 0.293 | – |
| | | | | (0.264–0.323) | – |
| Overall | 2,602,987 | 300,876,487 | 296 | 0.054 | 88.2 |
| | | | | (0.043–0.065) | (86.2–89.9) |
| 3mRNA | 236,895 | 23,626,638 | 16 | 0.069 | 74.2 |
| | | | | (0.028–0.110) | (54.8–85.2) |
| CC+mRNA | 840,136 | 89,423,892 | 76 | 0.064 | 77.0 |
| | | | | (0.048–0.080) | (69.7–83.0) |
| CCC | 137,002 | 19,206,580 | 37 | 0.053 | 75.0 |
| | | | | (0.032–0.074) | (63.4–82.9) |
| CCA | 1,393,752 | 203,137,349 | 210 | 0.037 | 86.3 |
| | | | | (0.030–0.044) | (83.2–88.8) |
| **Confirmed deaths** | | | | | |
| Unvaccinated | 749,188 | 137,223,419 | 1115 | 0.684 | – |
| | | | | (0.642–0.725) | – |
| Overall | 2,603,731 | 300,919,202 | 552 | 0.081 | 90.5 |
| | | | | (0.071–0.092) | (89.4–91.4) |
| 3mRNA | 236,930 | 23,629,947 | 9 | 0.146 | 87.7 |
| | | | | (0.010–0.282) | (76.1–93.7) |
| CC+mRNA | 840,425 | 89,441,770 | 169 | 0.133 | 81.0 |
| | | | | (0.112–0.154) | (76.8–84.0) |
| CCC | 137,122 | 19,214,510 | 98 | 0.115 | 79.3 |
| | | | | (0.092–0.138) | (73.8–83.7) |
| CCA | 1,394,118 | 203,161,303 | 372 | 0.059 | 90.8 |
| | | | | (0.052–0.066) | (89.4–92.0) |

*COVID-19 denotes coronavirus disease 2019. The Ministry of Health launched a COVID-19 vaccine first booster campaign on August 11, 2021, and a second booster campaign on February 14, 2022. The table shows the estimated vaccine effectiveness of mRNA-based second vaccine boosters for individuals with four different three-dose background regimes: (1) BNT162b2 primary series plus a homologous booster (3mRNA), a CoronaVac primary series plus (2) mRNA booster (CC+mRNA), (3) homologous booster (CCC), or (4) ChAdOx-1 booster (CCA), compared to unvaccinated individuals. Estimates were adjusted for time-varying vaccination exposure and clinical, demographic, and socioeconomic confounders at baseline (Supplementary Tables S1–S5).

(95% CI 63.4–82.9), and 86.3% (95%CI 83.2–88.8), for 3mRNA, CC+mRNA, CCC, and CCA, respectively. The adjusted vaccine effectiveness against COVID-19-related deaths was 87.7% (95% CI 76.1–93.7), 81.0% (95%CI 76.8–84.0), 79.3% (95%CI, 73.8–83.7), and 90.8% (95%CI 89.4–92.0) for 3mRNA, CC+mRNA, CCC, and CCA, respectively. Compared to the maximum observed in the cohort during the study period, vaccine effectiveness moderately decreased for all schemes and outcomes studied (Fig. 1), although most confidence intervals overlapped, with some exceptions (Supplementary Tables S6 and S7): vaccine effectiveness against COVID-19-related ICU admissions significantly decreased for the CCA scheme, and protection against COVID-19-related deaths significantly decreased for CC+mRNA, CCC, and CCA (Fig. 1 and Supplementary Tables S6 and S7).

## Discussion
Our cohort represents a population immunized with various schemes of COVID-19 vaccines. Ninety-one percent of the cohort received a primary series of CoronaVac, combined with a first booster based on a viral-vectored (53%) or an mRNA vaccine (37%). The remaining participants (9%) received a primary series and a first homologous booster of either CoronaVac (CCC) or BNT162b2 mRNA (3mRNA) vaccines. Similar to the overall effect of a second mRNA booster dose, the scheme-specific vaccine effectiveness of these boosters remained high by the end of the follow-up, ranging from 74.2% to 86.3% effectiveness against COVID-19-related ICU admission and 79.3% to 90.8% against

death. The CCA vaccination scheme plus a second mRNA booster dose provided the highest protection against COVID-19-related deaths. Our study was not designed to demonstrate the superiority of any specific vaccination scheme. However, our results suggest that mix-and-match strategies may provide comparable protection against severe disease to homologous, mRNA-based regimens.

Our findings are consistent with previous studies on second booster doses. A recent summary of studies on mRNA vaccines[25], including eight studies from Israel, one from Canada[21], and one from the USA[24], all conducted during the Omicron outbreak, suggests that a second mRNA booster enhances protection compared to three vaccine doses and unvaccinated individuals, particularly for severe outcomes. While not directly comparable, due to differences in study design, population characteristics, and outcome definitions, the studies in Canada[21] and the USA[24] estimate the vaccine effectiveness of a second mRNA booster compared to unvaccinated individuals. The study in Canada, conducted among residents in long-term care facilities aged 60 years or older, found that the effectiveness of a fourth dose against severe outcomes was 87% (95% CI 82–90)[21]. In the USA, research was conducted among adults aged 50 years or older without immunocompromising conditions. Those results showed that the effectiveness of a fourth mRNA dose against hospitalization was 80% (95% CI 71–85). These studies have a relatively short follow-up time of two to ten weeks after the fourth dose and thus do not fully capture immunity waning.

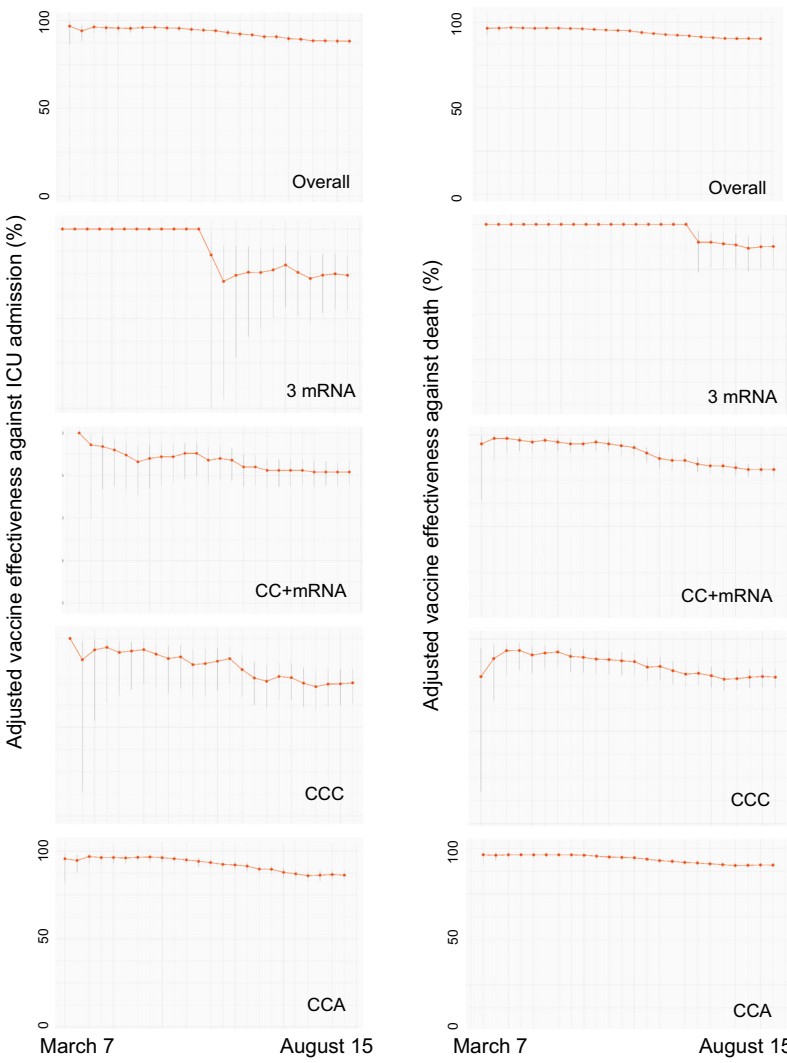

**Fig. 1 | Overall vaccine effectiveness against COVID-19-related ICU admissions and death of mRNA-based second vaccine boosters for each week, March 7 through August 15, 2022.** 3mRNA denotes individuals who received a BNT162b2 primary series plus a homologous first booster dose. CC+mRNA, CCC, and CCA denote individuals with a primary series of CoronaVac plus a first mRNA, homologous, and a ChAdOx-1 first booster dose, respectively. Vaccine effectiveness estimates are presented as point estimates with standard 95% Wald confidence intervals. The corresponding numerical values for vaccine effectiveness against COVID-19-related ICU admissions and deaths are shown in Supplementary Tables S6 and S7, respectively.

At least two other studies have examined the effectiveness of a second booster dose. A preprint study in Thailand examined the effectiveness of a second booster dose based on ChAdOx1 among adults with various three-dose background regimes, including inactivated vaccines[36]. The study found vaccine effectiveness against symptomatic COVID-19 of 73% (95%CI 48–89) but did not examine its effectiveness against severe outcomes. Using the same cohort, another study in Thailand found no severe outcomes for patients with a second booster dose based on ChAdOx1, BNT162b2, and mRNA-1273[26]. The assessment was conducted after an average of 53 days (interquartile range: 29–75 days) following their last dose. A study in Sweden examined all-cause mortality among fourth-dose recipients of BNT162b2 and mRNA-1273 compared to third-dose recipients and found a decrease in short-term risk of death of about 70% among individuals older than 80 years[27]. Similar to our findings, the study in Sweden also found moderate waning after 2 months[27]. Likewise, a study in Korea[31] and in Portugal[32] found that a fourth mRNA COVID-19 vaccine dose provided high protection against severe illness compared to a third dose in individuals older than 60 years. Last, anecdotal evidence from two participants in a cohort of healthcare workers in the USA suggests that a second homologous mRNA booster may induce a substantial and durable neutralizing antibody response[37].

Our study has some limitations. As an observational study, there is a risk of selection bias if the vaccinated and unvaccinated groups differ in a systematic way, such as risk aversion, which affects the probability of exposure to the treatment and the risk of SARS-CoV-2 infection. We adjusted our estimates with observable clinical, demographic, and socioeconomic confounders that could affect vaccination and the risk of infection, but there could be residual confounding. Second, our study design did not allow us to separate the effect of immunity waning and the emergence of new SARS-CoV-2 lineages with increased immune evasion on our vaccine effectiveness estimates against severe disease for a second booster dose based on mRNA vaccines. Our work aimed to provide policymakers with estimates of vaccine effectiveness for different populations and time periods. Through a prospective monitoring system, weekly reports on the vaccine effectiveness of booster shots were provided to the Ministry of Health. Considering comparability, we made the decision to focus on overall protection levels across different populations and stages of the vaccination campaign instead of the time elapsed since the last vaccine dose. This

choice was based on the understanding that vaccination campaigns are dynamic, featuring diverse intervals between doses, which poses a challenge in establishing a consistent timeframe to assess the decline of immunity. Third, COVID-19 is a notifiable disease in Chile. While RT-PCR and antigen tests are freely available for FONASA affiliates, it is possible that some individuals may have used an antigen test at home and not reported to the health system. Fourth, misclassification of asymptomatic or mildly symptomatic cases is possible, as they are less likely to seek healthcare. Unvaccinated individuals might be less likely to seek testing for mild COVID-19, leading to an underestimation of protection against symptomatic infection, while vaccinated individuals may develop mild symptoms due to vaccine-induced protection, potentially overestimating protection against symptomatic infection. The impact of these unmeasured confounders on the estimates is uncertain; however, it is unlikely to have affected effectiveness estimates for ICU admission and death related to COVID-19. Last, the Chilean Ministry of Health has limited genomic surveillance capabilities. The ministry's strategy has focused on detecting variants of concern through traveler and community surveillance but uses a non-probabilistic sampling strategy. Therefore, we do not have representative data to estimate the true prevalence of these variants or sublineages and their potential effect on vaccine effectiveness.

Overall, our study shows that a second booster dose with mRNA vaccine had high effectiveness against severe COVID-19, independent of the COVID-19 vaccine scheme received in the past. Overall protection against COVID-19-related ICU admission and death showed a moderate decrease of about 9% and 6%, respectively, by the end of the follow-up. The introduction of the BA.4 and BA.5 Omicron sub-lineages during the second part of the study period (Supplementary Fig. S3) and immunity waning due to a decline in the circulating levels of neutralizing antibodies as described for primary series and first booster doses may explain the decrease in protection[38–40]. We provide evidence to support mRNA-based second boosters following various background COVID-19 vaccine schemes, including widely used inactivated vaccines. These results suggest that a second booster would help minimize severe COVID-19, including deaths, and decrease the impact on the health system. The observed duration of protection for a second booster dose suggests there may be longer-acting protection after repeated immunization with both homologous and heterologous schemes and that no additional boosters are needed at six months.

# Methods
## Exposures and outcomes
The Ministry of Health in Chile requires that all suspected COVID-19 cases are notified to health authorities through an online platform and undergo confirmatory laboratory testing based on RT-PCR or antigen tests. This is the source for the COVID-19 case count in this study. We assessed the effectiveness of mRNA-based second vaccine boosters for individuals with four different three-dose background regimes: (1) BNT162b2 primary series plus a homologous booster (3mRNA), (2) CoronaVac primary series plus mRNA booster (CC+mRNA), (3) CoronaVac primary series plus homologous booster (CCC), and (4) CoronaVac primary series plus ChAdOx-1 booster (CCA). We estimated vaccine effectiveness weekly, from February 14, 2022, to August 15, 2022, against admission to an intensive care unit (ICU) and death (U07.1) associated with laboratory-confirmed SARS-CoV-2 infection.

We considered the onset of symptoms as a proxy for the time of infection. We used the time from the beginning of the second booster campaign on February 14, 2022, to the onset of symptoms of the event of interest, symptomatic cases that required hospitalization and symptomatic cases that died because of COVID-19 as the clinical endpoint. Participants were classified into unvaccinated and fully immunized individuals (≥14 days after receipt of the second booster dose). Individuals were excluded from the unvaccinated group when they received the first COVID-19 vaccine. We excluded the period between

the first COVID-19 vaccine dose and 13 days after the second booster dose from the at-risk person-time.

To understand the additional protection conferred by the second booster compared to the first booster, we also considered individuals with a three-dose regime as a comparison group. In this case, we considered the period 13 days after the first booster dose, regardless of the time of the booster uptake.

## Statistical analyses
Descriptive data was compared using Pearson's $\chi^2$ tests. We estimated hazard ratios using an extension of the Cox hazards model to account for the time-varying vaccination status of participants[8,35,41]. We estimated vaccine effectiveness using the hazard ratio between the treated and comparison group (i.e., non-treated individuals or individuals with a three-dose regime). We used a stratified version of the Cox hazards model[42] with time-dependent covariates to compare the risk of the event of interest between immunized (i.e., four doses) and non-immunized participants or participants with three doses at each event time. Under the stratified Cox model, each combination of predictors has a specific hazard function that can evolve independently. The model was stratified by age, sex, region of residence, income, nationality, and whether the patient had underlying conditions that have been associated with severe COVID-19, based on strata shown in Supplementary Tables S1–S5.

Let $T_i$ be the time-to-event of interest, from February 14, 2022, for the $i$-th individual in the cohort, $i = 1, \ldots, n$. Let $x_i, i = 1, \ldots, n$, be a $p$-dimensional vector of individual-specific characteristics, such as age and sex, and $z_i(t)$ be the time-dependent treatment indicator. The model assumes that the time-to-events are independent and with probability distribution given by

$$T_i | x_i, z_i \sim f(t, | x_i, z_i), i = 1, \ldots, n, \quad (1)$$

The time-to-event distribution is given by

$$f(t, | x_i, z_i) = \lambda_{x_i,0}(t) \exp\left\{\beta_{z_i(t)}\right\} \times \exp\left\{-\exp\left\{\beta_{z_i(t)}\right\} 0t \int \lambda_{x_i,0}(u) du\right\}, \quad (2)$$

with $\beta_k \in \mathbb{R}$ being the regression coefficient measuring the effectiveness of the $k^{th}$ treatment, and $\lambda_{x,0}$ is the predictor-specific baseline hazard function,

$$\lambda_{x_i,0}(t) = \lim_{h \to 0}\left\{\frac{P_{x_i,0}(t \le T \le t+h | T \ge t)}{h}\right\}, \quad (3)$$

where $P_{x_i,0}$ is the baseline probability distribution of the time-to-event in each strata.

We estimated the vaccine effectiveness as $100\% \cdot (1 - \exp\{\beta_k\})$. We show the adjusted vaccine effectiveness results, including covariates as controls (age, sex, region of residence, nationality, income, and comorbidities). We computed standard 95% Wald confidence intervals (95% CI) for the estimates. Inference was based on a partial likelihood approach[43]. Each term in the partial likelihood of the effectiveness regression coefficient corresponds to the conditional probability of an individual expressing the outcome of interest from the risk set at a given calendar time.

We analyzed the data with the survival package[44] of R, version 4.0.5[45].

## Model assumptions
The stratified version of the extended Cox proportional-hazards model assumes: (1) that the units in each stratum have a different baseline hazard rate, which depends on covariates shown in Supplementary

Tables S1–S5 (i.e., age, sex, region of residence, income, nationality, and whether the patient had underlying conditions that have been associated with severe COVID-19), (2) the coefficients for all these covariates do not change with time, (3) vaccination status changes with time, and (4) vaccine effectiveness is constant throughout the period considered for each estimate.

### Ethics statement
The Comité Ético Científico Clínica Alemana Universidad del Desarrollo, Santiago, Chile, gave ethical approval for this research. The study was deemed exempt from informed consent because it presented no potential harm to the participants, guaranteed the safeguarding of their privacy, relied on data from mandatory surveillance provided by the Ministry of Health, and provided crucial information for the management of the COVID-19 pandemic.

### Reporting summary
Further information on research design is available in the Nature Portfolio Reporting Summary linked to this article.

## Data availability
Due to data privacy regulations in Chile, individual-level data from this study cannot be disclosed, as mandated by Law No. 19.628. However, aggregated data related to vaccination, including demographic information and COVID-19 incidence, is publicly accessible at https://github.com/MinCiencia/Datos-Covid19/. The Transparency Law (Law No. 20.285) recognizes the right of all individuals to access public information. More detailed data regarding COVID-19 vaccinations and demographics can be requested directly from the Ministry of Health through the provided link. It is important to note, however, that the right to information is not without limits, as the law designates certain sensitive information as reserved. Public organizations have a period of 20 business days to respond to requests. For additional information, inquiries can be directed to transparencia@minsal.cl, and specific requests can be submitted using the following link: http://transparencia.redsalud.gob.cl/transparencia/public/ssp/solicitud_informacion.html.

## Code availability
The code for preparing and analyzing data is available upon request from the corresponding author.

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

## Acknowledgements

This research was supported by the Agencia Nacional de Investigación y Desarrollo (ANID) through the Fondo Nacional de Desarrollo Científico y Tecnológico (FONDECYT) grant N° 1220907 to A.J.; Proyecto Anillo grant N° ATE220061, ANID, and Advanced Center for Chronic Diseases (ACCDiS) ANID FONDAP grant N° 15130011 to R.A. ; and Research Center for Integrated Disaster Risk Management (CIGIDEN) ANID FONDAP grant N° 1522A0005 and The Canadian Institute for Advanced Research CIFAR under the Humans and the Microbiome programme to E.U. The funders of this study had no role in the study design, in the collection, analysis, and interpretation of data, in the writing of this manuscript or in the decision to submit the article for consideration for publication.

## Author contributions

A.J., C.C., and R.A. conceived and designed the study. A.J., C.C., C.G., M.N., and R.A. managed and analyzed the data. A.J., E.U., R.A. wrote the first draft of the manuscript. A.J., C.C., E.U., C.G., M.N., M.B., J.F., and R.A. critically reviewed and edited the manuscript. V.V., H.G.-E., and C.C. had access to vaccine safety data. All the authors are responsible for the study design, data collection, and data analysis. All authors have read and approved the final version of the manuscript. The authors vouch for the accuracy and completeness of the data and accept responsibility for publication. A.J., C.C. and R.A. contributed equally to the manuscript.

## Competing interests

R.A. has received consulting fees from AstraZeneca. R.A. and A.J. have received consulting fees from Pfizer and research support from Sinovac. This support is not related to this article. The remaining authors declare no competing interests.
