## [Peer Review File · Nature Communications]

Effectiveness of the second COVID-19 booster against Omicron: a large-scale cohort study in ChileREVIEWER COMMENTS

Reviewer #1 (Remarks to the Author):

This paper used national surveillance data to assess VE of 4 different vaccine schedules. Overall, the results showed at a second booster dose increased protection against severe disease, including ICU admission and death.

These are important data; though many estimates of VE during BA.4/5 have been published, this paper includes comparisons of different schedules and vaccines used in middle- and low-income countries. I'm not clear on why the authors could not evaluate time since most recent dose to look at waning (e.g., VE by month since last dose) and think that would add valuable information.

I had a number of comments throughout:

Line 50 - Not sure what this sentence means.

Line 82 – Curious to know more about how prior infection is captured. Is home testing used in Chile?

Line 83 – Were immunocompromised individuals included in this analysis? Do they have the same schedule as everyone else?

Line 99/100 – moderate but significant reduction in VE compared to what? Pre-Delta?

Line 103 – helpful to keep results in same order as methods.

Line 106 – again decreased compared to what?

Line 111-114 – these sentences make it sound like all individuals got CoronaVac as the primary series, but you have estimates for 3 mRNA vaccines.

Line 131 – I'm not sure what this is referencing. Reference #15 is immunocompetent individuals.

Line 141/limitations - How well is prior infection measured? Also assume you don't have sequencing on all cases, so another limitation is using time to determine variant.

Table 1 – would be helpful to add median time since last dose to these rows.

Reviewer #2 (Remarks to the Author):

Authors report a vaccine effectiveness study of a large national cohort in Chile, with a focus on a second booster (fourth) vaccine dose relative to zero doses. The strengths of the analysis include the large cohort size in the context of a national database, as well as the ability to stratify by different strategies for the first 3 vaccine doses. There have been relatively few studies assessing the role of boosters (and second boosters) in the context of an initial inactivated vaccine strategy, and therefore this analysis is in general a welcome addition to the covid vaccine effectiveness literature. Overall it is reassuring to see that the varying effectiveness estimates by vaccine product fit with what is generally known about the relative effectiveness from placebo-controlled trials. Unfortunately, observational studies like this can be very vulnerable to biases inherent to the study design, along with changes in immunity, which the authors partially highlight in their discussion. I have some comments and suggestion for the authors, below.

1. The paper assesses the relative effectiveness of the second booster dose compared to unvaccinated individuals. Is this really the comparison of greatest interest? One could argue that we are really interested in the relative effectiveness compared to three doses. At minimum it would be helpful to see three-dose effectiveness relative to the unvaccinated as a benchmark.
2. I am somewhat concerned about the strategy used to generate the cohort. If I understand correctly: first, the authors consider only those who were unvaccinated or were eligible for a first booster dose as of August 2021; second they excluded anyone who had a positive test up to February 2022. Why not include people who were not previously infected, and eligible for the fourth dose at the time the second booster campaign began (or unvaccinated)?
3. At this stage of the pandemic, one of the principal sources of bias in a study like this, which uses routinely collected clinical data (in other words, testing is not systematic but instead is driven by participants seeking it out), is misclassification of previously infected (but not test-confirmed), unvaccinated people as never-infected unvaccinated. It's highly plausible that people who are unvaccinated at this point (particularly in a population with very high vaccine coverage like Chile) are also unlikely to seek testing for milder illness, and therefore misclassification would be differential between the unvaccinated and vaccinated groups. This is a particular concern with a study period that starts after the Delta and

Omicron waves, during which rates of infection among the unvaccinated were very high. This issue will lead to underestimates of effectiveness (and in some cases a false sense of immune waning over time), because the unvaccinated group may actually have a substantial degree of immunity from prior undocumented infection(s). Two suggestions to address this. First, it would be helpful to know how many unvaccinated people in the overall cohort were excluded for prior documented infection (i.e., we would expect the 757,726 to be a very small subset). Second, one way to try to mitigate this bias would be to control for baseline testing behavior prior to the study period in the models.

4. How were participants who became infected during the study period but not hospitalized considered in the analysis after that time point?

5. Lines 44-48: I'm not sure that I agree with this framing. As far as I'm aware, there is one placebo-controlled RCT of a third vaccine dose that assesses a clinical outcome (Moreira et al, NEJM 2022), showing reduction in symptomatic infection risk relative to two doses. This is the strongest pre-Omicron justification for addition of a third vaccine dose, so at minimum would recommend including this reference. Prior to the emergence of omicron, the severe immune waning suggested in various observational settings was not borne out to the same extent in randomized trials with long-term follow up (see <https://www.fda.gov/media/152176/download> page 21; Baden et al NEJM letter December 2021).

6. Line 50-52: This statement is unclear – there was loss in protection for both primary series and booster doses in the context of omicron, though booster doses restored a substantial amount of protection, making the relative importance of booster (third) doses much greater in the context of omicron.

7. Line 111-114: What about those who received a primary mRNA series?

8. Extended figure 1: This figure is unclear. 757,726 were not vaccinated by the end of the follow-up period, but the inclusion criteria in the first box suggests needing to have received at least two vaccine doses?

Reviewer #3 (Remarks to the Author):

This article explores an important topic of the second booster mRNA vaccine effectiveness (VE) against COVID-19-related ICU admissions and deaths during the Omicron circulation in a large population cohort in Chile. The authors estimate VE for different three-dose background regimes and also evaluate the change in vaccine-induced protection over time. The statistical methods look appropriate and overall the manuscript is good and addresses a relevant topic but there are some points that the authors may be willing to address.

1. The title should be more informative including the time, place, and design of the study.

2. Authors use the terms “an additional booster” and “second booster dose” in the manuscript, please standardize the terminology.

3. The manuscript will benefit from a more detailed description of the epidemiological context and the second booster vaccination campaign in Chile. For instance, were there any non-pharmacological interventions in place during the study period?

Regarding the vaccination campaign, the authors indicated that it prioritized the elderly and immunocompromised individuals and that vaccination roll-out has been described in more detail elsewhere, indicating reference 19. Reference 19 is Jara, A. et al. Effectiveness of an Inactivated SARS-CoV-2 Vaccine in Chile. *N Engl J Med* 385, 875-884, doi:10.1056/NEJMoa2107715 (2021)., that only describes the primary vaccination roll-out. Recommendations for the second booster dose vary considerably by country. Please clearly identify the population eligible for the second booster dose uptake in Chile and briefly describe the second booster vaccination campaign roll-out.

4. The authors reported the second booster vaccine coverage reached by August 29, 2022. If the vaccine coverage has not been estimated with your data please add the reference that supports the estimate.

5. The study period is not consistently reported in the manuscript, some times it is February 14 to August 15, 2022 others February 14, 2022, to August 18, 2022. Please check the study period in the text and the tables.

6. The description of the inclusion and exclusion criteria needs to be clarified and the exclusion flowchart may be improved. What was the recommended time interval between the first and the second booster? Please add this information to the inclusion criteria.

7. There are some discrepancies between the flowchart and its description in the main text.

In the text, the authors indicated “study cohort included adults aged ≥ 20 years and affiliated with the Fondo Nacional de Salud (FONASA), the public national healthcare system, who completed a CoronaVac or BNT162b2’s two-dose primary series at least 120 days before the beginning of the follow-up on August 11, 2021, when the first-booster campaign was launched, and unvaccinated individuals “In the flowchart, the description is different, ...” at least 120 days before August 8, 2021” Please check the dates. Also, please check the number of vaccinated in the flowchart, it is different from the one previously reported in the text.

8. In the footnote to the flowchart, the authors stand that “1,788 persons received three doses of CoronaVac plus a homologous CoronaVac second booster. We did not include the latter in the study due to the small sample size”. If individuals were excluded from all analyses, it should be mentioned in the exclusion criteria in the main text as well. Please clarify this point.

9. The authors stand that they excluded individuals with probable or confirmed COVID-19 reported before February 15, 2022. How the “probable” COVID-19 case is defined? Please clarify this point and add the information on the number of excluded due to the previous infection to the flowchart - Extended Fig.1.

10. The description of variables included in the Cox model is not consistent First, the authors said: “Hazard ratios were adjusted for age, sex, region of residence, nationality, income, and comorbidities conditions (Extended Data Tables 1-5)”. Later the authors mentioned that “We show the adjusted vaccine effectiveness results, including covariates as controls (age, gender, region, nationality, health insurance category, and comorbidities)”. Please clarify if the income or the health insurance category or both were included in the model.

In the text the authors mention variable age (that is usually interpreted as age in years), in tables, the authors present the variable Age group. Please check this point.

Also, the authors named the region of residence as one of the covariates under study. Later, the authors renamed this covariate as a Cohort location. Please standardize the nomenclature in the text and the tables.

11. In the statistical analysis description, it is not clear which variables were used as a starta in the Cox models and how Cox model assumptions were checked. Please add this information to the online methods sections.

12. To estimate VE, the authors used unvaccinated as a reference category. It would be also

important to understand the additional protection conferred by the second booster dose compared to the first booster. Please consider reporting relative VE using the first booster as a reference category.

13. The authors assessed change in VE over time, estimating VE on a weekly basis. However, the weekly number of COVID-19-related severe outcomes (ICU admissions and deaths) is low, in particular when specific regimens are considered, leading to a lack of precision. Confidence intervals may be quite wide (for instance VE=85%, CI95%:0.0-98.0), as the authors documented in the study limitations. The assessment of change in VE over time is important and without a doubt can be one of the study's strengths. The authors should consider to change the time window, a month instead of a week will be more appropriate to estimate VE for severe outcomes and it will improve the estimates precision.

Response to referees' comments

Nature Communications NCOMMS-22-40863-T

Effectiveness and duration of a second COVID-19 vaccine booster: a large-scale prospective cohort study during the SARS-CoV-2 Omicron outbreak in Chile

Date: May 23 10, 2023

We are grateful for the thoughtful comments and suggestions from the three referees. We think our manuscript has substantially improved as a result of this revision. The referees' comments are numbered consecutively, and our responses are labeled as "Response" and **highlighted in yellow**. New references included in our responses are listed at the end of this document. In addition to the comments noted below, we made minor editorial and formatting updates to the manuscript's main text and updated a few references.

Reviewer #1

Remarks to the author

R1.1 This paper used national surveillance data to assess VE of 4 different vaccine schedules. Overall, the results showed at a second booster dose increased protection against severe disease, including ICU admission and death.

These are important data; though many estimates of VE during BA.4/5 have been published, this paper includes comparisons of different schedules and vaccines used in middle- and low-income countries. I'm not clear on why the authors could not evaluate time since most recent dose to look at waning (e.g., VE by month since last dose) and think that would add valuable information.

Response: Thank you for your feedback. We appreciate your suggestion to consider the time since the last vaccine dose in assessing the waning of immunity. It is indeed an important aspect to

consider; however, there are specific reasons why we did not incorporate this element in our current study. The main aim of our study was to generate summary statistics of vaccine effectiveness that policymakers could use in tracking the overarching protection levels within various populations at different stages in time. We sought to create a broad perspective of vaccine efficacy, spanning various vaccination schedules and vaccine types, with a focus on low- and middle-income countries. Please recall that this study was done in collaboration with the Ministry of Health and in the context of urgent policymaking. An analysis based on the time elapsed since the last vaccine dose may yield valuable insights into immunity's potential decay over time. However, this additional layer would also introduce an element of complexity that could constrain the comparability of our findings across distinct populations and different time frames. The reality of vaccination campaigns is their fluidity, coupled with the fluctuating intervals between vaccine doses across different regions. This dynamic makes it complex to delineate a consistent time frame to evaluate the decline of immunity. To address the reviewer's concern, we now explain these reasons explicitly in the manuscript (p.8, lines 187-191).

I had a number of comments throughout:

R1.2 Line 50 - Not sure what this sentence means.

Response: Thank you for bringing this to our attention. We think the referee is referring to the sentence "As happened with the primary series of COVID-19 vaccines, research showed substantial reductions in protection against symptomatic COVID-19". We now realize that the sentence may have lacked adequate context and have rephrased it for clarity:

P.3, lines 52-57: "The emergence and spread of the B.1.1.529 Omicron variant of SARS-CoV-2, and its sub-lineages, caused an unprecedented global surge in COVID-19 cases.^{1,2} This raise may be partly explained by Omicron's high transmissibility and ability to partially evade natural and vaccine-induced immunity against SARS-CoV-2,^{3,4} and from the waning of vaccine protection.⁵⁻⁷

While booster doses restored a substantial amount of protection,⁸⁻¹¹ further research showed substantial reductions in the protection of boosters against symptomatic COVID-19,¹²⁻¹⁴ as happened with the primary series of COVID-19 vaccines.”

R1.3 Line 82 – Curious to know more about how prior infection is captured. Is home testing used in Chile?

Response: Covid-19 is a nationally notifiable disease in Chile. Healthcare providers are required by law to notify health authorities of all suspected Covid-19 cases through Epivigila, an electronic surveillance system, and undergo laboratory confirmation. Chile has a hybrid public-private health system, including insurance and service provisions. Healthcare coverage is high; about 98% of individuals in Chile have health insurance, with about 80% of the population affiliated to FONASA, the public national healthcare system. In our study, all Covid-19 cases were laboratory-confirmed SARS-CoV-2 infections based on RT-PCR or antigen test and corresponded to ICD-10 code U07.1. RT-PCR and antigen tests are freely available for FONASA affiliates in healthcare centers throughout the country in a network of primary care centers and referral hospitals. FONASA does not discriminate by age, gender, income, number of dependents, pre-existing conditions, or nationality. If a person had symptoms but had no confirmed SARS-CoV-2 infection (by PCR or antigen test), then the person was not considered as Covid-19 following our case definition and would not have appeared as Covid-19 in our surveillance dataset. As stated in the main text, we excluded individuals with probable or confirmed COVID-19 according to reverse-transcription polymerase-chain-reaction assay for SARS-CoV-2 or antigen test reported before February 15, 2022. While it is possible that some cases could have used an antigen test at home and not reported to the healthcare system, these cases are probably negligible due to the relatively high costs of the test compared to freely available tests in healthcare centers across the country. To

address the reviewer's comment, we have clarified this in the manuscript (p.4, lines 83-92), and noted this limitation in the discussion (also see R1.10).

R1.4 Line 83 – Were immunocompromised individuals included in this analysis? Do they have the same schedule as everyone else?

Response: The reviewer is correct; immunocompromised individuals were included in the analysis. Our vaccine effectiveness estimates adjusted for age, sex, region of residence, income, nationality, and whether the patient had underlying conditions associated with severe Covid-19 illness (Extended Data Tables 1-5). The vaccination campaigns, including boosters, prioritized older adults, persons with underlying conditions, and front-line workers. We have described the Covid-19 vaccination campaigns and the Chilean healthcare system in previous articles.^{8,15} We now clarify this in the main text (p.5, lines 90-92).

R1.5 Line 99/100 – moderate but significant reduction in VE compared to what? Pre-Delta?

Response: Thank you for bringing this to our attention. The moderate but significant reduction in vaccine effectiveness is compared to the maximum vaccine effectiveness against COVID-19-related ICU admissions and deaths observed in the cohort, as shown in Fig.1 and Extended Data Tables 6-7. We have now clarified this in the main text: P.6, lines 122-126, "These estimates represent a moderate but significant reduction in vaccine effectiveness compared to the maximum observed in the cohort during the study period of 96.8% (95% CI 86.8-99.2) for the prevention of ICU admission and 96.7% (95% CI 93.0-98.3) for the prevention of COVID-19 related death (Fig.1 and Extended Data Tables 6-7).

R1.6 Line 103 – helpful to keep results in the same order as methods.

Response: Thank you, we have rearranged the results accordingly.

R1.7 Line 106 – again decreased compared to what?

Response: We have now clarified that we meant compared to the maximum observed in the cohort during the study period (please see R1.5).

R1.8 Line 111-114 – these sentences make it sound like all individuals got CoronaVac as the primary series, but you have estimates for three mRNA vaccines.

Response: Thank you for noting this error; we meant homologous boosters with CoronaVac (CCC) or BNT162b2 mRNA (3mRNA) vaccines. We have clarified this in the text (p.7, lines 145-147).

R1.9 Line 131 – I'm not sure what this is referencing. Reference #15 is immunocompetent individuals.

Response: Thank you for noting this error. We meant that Link-Gelles et al.¹⁶'s research was conducted among adults aged 50 years or older without immunocompromising conditions. We have corrected this error in the manuscript.

R1.10 Line 141/limitations - How well is prior infection measured? Also assume you don't have sequencing on all cases, so another limitation is using time to determine variant.

Response: Thank you for these suggestions. The reviewer notes two potential limitations. First, the possibility that not all symptomatic SARS-CoV-2 infections are notified. Second, not all cases are sequenced. We address these separately.

First, as mentioned in our response to R1.3, healthcare providers are required by law to notify health authorities of all suspect Covid-19 cases through EpiVigila, an electronic surveillance

system, and undergo laboratory confirmation. Covid-19 cases were laboratory-confirmed SARS-CoV-2 infections based on RT-PCR or antigen test and corresponded to ICD-10 code U07.1. RT-PCR and antigen tests are freely available for FONASA affiliates in healthcare centers throughout the country in a network of primary care centers and referral hospitals. While it is possible that some cases could have used an antigen test at home and not reported to the healthcare system, these cases are probably negligible due to the relatively high costs of the test compared to freely available tests in healthcare centers across the country.

Second, the reviewer correctly underscores that the Chilean Ministry of Health has limited genomic surveillance capabilities. The ministry's strategy has focused on detecting variants of concern through traveler and community surveillance but uses a non-probabilistic sampling strategy. The manuscript reports the estimated distribution of the predominant SARS-CoV-2 lineages in Chile between February and September 2022 based on data shared on the GISAID platform (n=12,578).

We do not have representative data to estimate the true prevalence of these variants and their potential effect on vaccine effectiveness. Our study focuses on the vaccine effectiveness for various vaccine schedules, but we do not provide estimates for specific variants or sub-lineages. Based on Omicron's characteristics and the observed change in samples received, we are confident that Omicron was the predominant variant during the study period.

We added these limitations to the discussion to address the reviewer's comment (p.8, lines 179-186).

R1.11 Table 1 – would be helpful to add median time since last dose to these rows.

Response: Thank you for the suggestion. Please see our response to R1.1.

Reviewer #2

R2.1 Authors report a vaccine effectiveness study of a large national cohort in Chile, with a focus on a second booster (fourth) vaccine dose relative to zero doses. The strengths of the analysis include the large cohort size in the context of a national database, as well as the ability to stratify by different strategies for the first 3 vaccine doses. There have been relatively few studies assessing the role of boosters (and second boosters) in the context of an initial inactivated vaccine strategy, and therefore this analysis is in general a welcome addition to the covid vaccine effectiveness literature. Overall it is reassuring to see that the varying effectiveness estimates by vaccine product fit with what is generally known about the relative effectiveness from placebo-controlled trials. Unfortunately, observational studies like this can be very vulnerable to biases inherent to the study design, along with changes in immunity, which the authors partially highlight in their discussion. I have some comments and suggestion for the authors, below.

Response: Thank you for the thorough review and helpful comments.

R2.2 The paper assesses the relative effectiveness of the second booster dose compared to unvaccinated individuals. Is this really the comparison of greatest interest? One could argue that we are really interested in the relative effectiveness compared to three doses. At minimum it would be helpful to see three-dose effectiveness relative to the unvaccinated as a benchmark.

Response: Thank you for the suggestion. We have now included Extended Data Table 8, showing the overall weekly effectiveness against ICU admissions and death for adults >20 years of mRNA-based second vaccine boosters compared to three-dose background regimes in Chile, March 7, 2022, through August 15, 2022. We have also included a graph showing these data (Extended Data Fig. 4). The table and the figure (see below) show that a second vaccine booster effectively prevents COVID-19-related ICU admission and death and provides additional protection compared to the first booster. We also added this clarification on p.6, lines 125-131.

Extended Data Table 8. Overall weekly vaccine effectiveness against ICU admissions and death for adults >20 years of mRNA-based second vaccine boosters compared to three-dose background regimes in Chile, March 7, 2022, through August 15, 2022

Report date	ICU admissions		Death	
	VE (%)	95% CI	VE (%)	95% CI
March 7	90.9	(61.3-97.9)	93.8	(87.3-96.9)
March 14	78.1	(53.5-89.7)	92.9	(88.4-95.6)
March 21	87.2	(74.2-93.7)	92.9	(89.3-95.3)
March 28	87.5	(77.5-93.0)	91.7	(88.1-94.2)
April 4	85.4	(75.5-91.3)	91.2	(87.8-93.6)
April 11	85.8	(77.1-91.2)	91.0	(87.8-93.4)
April 18	86.8	(78.8-91.8)	90.7	(87.4-93.1)
April 25	85.1	(76.3-90.6)	89.7	(86.3-92.2)
May 2	83.2	(73.8-89.2)	88.4	(84.8-91.2)
May 9	82.4	(73.2-88.5)	86.9	(83.1-89.9)
May 16	77.9	(67.2-85.1)	85.2	(81.1-88.4)
May 23	76.6	(66.0-83.8)	83.8	(79.6-87.1)
May 30	74.0	(63.1-81.6)	82.8	(78.6-86.2)
June 6	68.9	(57.0-77.5)	80.0	(75.6-83.6)
June 13	68.1	(57.0-76.4)	78.7	(74.4-82.3)
June 20	68.8	(58.8-76.4)	77.2	(73.0-80.8)
June 27	67.4	(57.7-74.9)	76.3	(72.2-79.9)
July 4	68.4	(59.6-75.3)	75.5	(71.5-79.0)
July 11	65.3	(56.1-72.6)	73.9	(69.8-77.3)
July 18	65.2	(56.4-72.3)	72.8	(68.8-76.3)
July 25	62.8	(53.8-70.1)	71.6	(67.6-75.1)
August 01	62.0	(52.9-69.3)	70.8	(66.8-74.4)
August 08	61.1	(52.1-68.5)	70.6	(66.6-74.2)
August 15	60.9	(51.9-68.2)	70.1	(66.1-73.7)

*COVID-19 denotes coronavirus disease 2019. The Ministry of Health launched a COVID-19 vaccine first booster campaign on August 11, 2021, and a second booster campaign on February 14, 2022. Estimates were adjusted for time-varying vaccination exposure and clinical, demographic, and socioeconomic confounders at baseline (Extended Data Tables 1-5)

Extended Data Fig.4. Overall weekly vaccine effectiveness against COVID-19-related (A) ICU admissions and (B) death for adults >20 years of mRNA-based second vaccine boosters compared to three-dose background regimes in Chile, March 7, 2022, through August 15, 2022

R2.3. I am somewhat concerned about the strategy used to generate the cohort. If I understand correctly: first, the authors consider only those who were unvaccinated or were eligible for a first booster dose as of August 2021; second they excluded anyone who had a positive test up to February 2022. Why not include

people who were not previously infected, and eligible for the fourth dose at the time the second booster campaign began (or unvaccinated)?

Response: As the reviewer noticed, our cohort began on August 11, 2021, when the booster campaign was launched in Chile (the first booster shot campaign). The initial aim was to report the vaccine effectiveness of the first booster doses (see Jara et al.⁸). As the population eligible for a second booster was the same population eligible -and receiving- a first booster shot, we kept the date when the original cohort was created. However, we noticed this was unclear in our original submitted manuscript. Where we said, “We excluded individuals with probable or confirmed COVID-19 according to reverse-transcription polymerase-chain-reaction assay for SARS-CoV-2 or antigen test reported before February 15, 2022,” we should have said instead “(...) or antigen test reported before August 11, 2021”.

We have clarified the reviewer’s comment in the main text (p. 5, lines 103-106) and updated Extended Data Fig.1 accordingly to address the reviewer's comment.

R2.4. At this stage of the pandemic, one of the principal sources of bias in a study like this, which uses routinely collected clinical data (in other words, testing is not systematic but instead is driven by participants seeking it out), is misclassification of previously infected (but not test-confirmed), unvaccinated people as never-infected unvaccinated. It’s highly plausible that people who are unvaccinated at this point (particularly in a population with very high vaccine coverage like Chile) are also unlikely to seek testing for milder illness, and therefore misclassification would be differential between the unvaccinated and vaccinated groups. This is a particular concern with a study period that starts after the Delta and Omicron waves, during which infection rates among the unvaccinated were very high. This issue will lead to underestimates of effectiveness (and in some cases a false sense of immune waning over time), because the unvaccinated group may actually have a substantial degree of immunity from prior undocumented infection(s). Two suggestions to address this. First, it would be helpful to know

how many unvaccinated people in the overall cohort were excluded for prior documented infection (i.e., we would expect the 757,726 to be a very small subset). Second, one way to try to mitigate this bias would be to control for baseline testing behavior prior to the study period in the models.

Response: We agree that misclassifying previously infected individuals is a relevant concern. Because laboratory-confirmed Covid-19 cases depend on the patients' healthcare-seeking behavior, it is possible that asymptomatic or mildly symptomatic cases were missed in our study. As the reviewer suggests, it is theoretically possible that unvaccinated individuals are less likely to seek testing for mild manifestations of Covid-19, which would result in a systematic difference in laboratory-confirmed mild Covid-19 between vaccinated and unvaccinated groups. This would lead to an underestimation of protection against symptomatic infection. On the other hand, immunized individuals may be more likely to develop mild symptoms due to vaccine-induced protection than unvaccinated individuals. This would lead to an overestimated protection against symptomatic infection. Unfortunately, we do not have this type of data on our cohort or know of any studies that have characterized a representative sample of these groups in Chile. We cannot be sure in which direction, if any, these unmeasured confounders could potentially affect our estimates. However, this potential misclassification bias is unlikely to have impacted our effectiveness estimates for protection against Covid-19-related ICU admission and death. To address the reviewer's comment, we have now expanded this discussion in the limitations section of the manuscript (p.9, lines 193-199).

R2.5. How were participants who became infected during the study period but not hospitalized considered in the analysis after that time point?

Response: We agree with the reviewer that this is an important point. When ICU admission or death is the clinical endpoint considered, an individual in the cohort can potentially move through one or more disease states (e.g., mild symptoms to illness requiring hospitalization). Under the stratified

version of the Cox model, each combination of predictors has a specific hazard function that can evolve independently. We considered the time from the beginning of the follow-up to the onset of symptoms of the event of interest: symptomatic cases that required hospitalization and symptomatic patients that died because of Covid-19. We have clarified this in the methods (p.15, lines 336-342).

R2.6. Lines 44-48: I'm not sure that I agree with this framing. As far as I'm aware, there is one placebo-controlled RCT of a third vaccine dose that assesses a clinical outcome (Moreira et al, NEJM 2022), showing reduction in symptomatic infection risk relative to two doses. This is the strongest pre-Omicron justification for addition of a third vaccine dose, so at minimum would recommend including this reference. Prior to the emergence of omicron, the severe immune waning suggested in various observational settings was not borne out to the same extent in randomized trials with long-term follow up (see <https://www.fda.gov/media/152176/download> page 21; Baden et al NEJM letter December 2021).

Response: Thank you for bringing this research to our attention. To address the reviewer's concern, we have rephrased this statement added evidence by Moreira et al.¹⁷ and Baden et al.¹⁸, and toned down the statement about immunity waning, as we agree that existing evidence was only suggestive at that time. Please see p.3, lines 52-57.

R2.7. Line 50-52: This statement is unclear – there was a loss in protection for both primary series and booster doses in the context of omicron, though booster doses restored a substantial amount of protection, making the relative importance of booster (third) doses much greater in the context of omicron.

Response: Thank you for noting this; we agree with the reviewer and have clarified this statement in the main text (please see p.3, lines 52-57, and our response to R1.2).

R2.8. Line 111-114: What about those who received a primary mRNA series?

Response: Thank you for noting this error; we meant homologous boosters with either CoronaVac (CCC) or BNT162b2 mRNA (3mRNA) vaccines. We have clarified this in the text (p.7, lines 146-148).

R2.9. Extended figure 1: This figure is unclear. 757,726 were not vaccinated by the end of the follow-up period, but the inclusion criteria in the first box suggests needing to have received at least two vaccine doses?

Response: Thank you for noting this mistake. As we mention on p.4, lines 99-106, our study cohort included adults aged ≥ 20 years and affiliated with the Fondo Nacional de Salud (FONASA), the public national healthcare system, who completed a CoronaVac or BNT162b2's two-dose primary series at least 120 days before the beginning of the follow-up on August 11, 2021, when the first-booster campaign was launched, and unvaccinated individuals. We have now corrected the figure.

Reviewer #3

R3.1 This article explores an important topic of the second booster mRNA vaccine effectiveness (VE) against COVID-19-related ICU admissions and deaths during the Omicron circulation in a large population cohort in Chile. The authors estimate VE for different three-dose background regimes and also evaluate the change in vaccine-induced protection over time. The statistical methods look appropriate and overall the manuscript is good and addresses a relevant topic but there are some points that the authors may be willing to address.

Response: Thank you for the encouraging comments and for your thorough review.

R3.2. The title should be more informative including the time, place, and design of the study.

Response: Thank you for the suggestion. The new title is “Effectiveness and duration of a second COVID-19 vaccine booster: a large-scale prospective cohort study during the SARS-CoV-2 Omicron outbreak in Chile”

R3.3. Authors use the terms “an additional booster” and “second booster dose” in the manuscript, please standardize the terminology.

Response: Done.

R3.4. The manuscript will benefit from a more detailed description of the epidemiological context and the second booster vaccination campaign in Chile. For instance, were there any non-pharmacological interventions in place during the study period?

Response: Thank you for the suggestion. We have now expanded the description of the epidemiological context and the vaccination campaign in Chile in the main text (please also see R3.5). Most non-pharmaceutical interventions to control Covid-19 were no longer enforced, although health authorities still recommended washing hands frequently, adequate ventilation, and physical distancing in public places. Confirmed Covid-19 cases were required to stay at home, and facemasks were mandatory in public transport and other public spaces throughout the study period. We have added this additional context to the manuscript. Please see pp.4-5, lines 80-98.

R3.5 Regarding the vaccination campaign, the authors indicated that it prioritized the elderly and immunocompromised individuals and that vaccination roll-out has been described in more detail elsewhere, indicating reference 19. Reference 19 is Jara, A. et al. Effectiveness of an Inactivated SARS-CoV-2 Vaccine in Chile. *N Engl J Med* 385, 875-884, doi:10.1056/NEJMoa2107715 (2021), that only describes the primary vaccination roll-out. Recommendations for the second booster dose vary

considerably by country. Please clearly identify the population eligible for the second booster dose uptake in Chile and briefly describe the second booster vaccination campaign roll-out.

Response: Thank you for the suggestion. The Chilean Ministry of Health began the second booster campaign on February 14, 2022. All adults were eligible for a second booster dose, although priority was given to the elderly, front-line health workers, immunocompromised individuals, and persons with underlying conditions associated with the risk of severe COVID-19. The government tracks vaccination schedules through an electronic national immunization registry, and vaccination rollout was organized using a publicly available national vaccination schedule that assigns specific vaccination dates to eligible groups. Eligible individuals show up at their nearest vaccination site (e.g., primary healthcare clinic) with IDs; they do not need an appointment. A minimum of 20 weeks were required between the first and the second booster dose. We added this response's most salient parts in the main manuscript. Please see pp.4-5, lines 80-98.

R3.6 The authors reported the second booster vaccine coverage reached by August 29, 2022. If the vaccine coverage has not been estimated with your data please add the reference that supports the estimate.

Response: Done.

R3.7. The study period is not consistently reported in the manuscript, sometimes it is February 14 to August 15, 2022 others February 14, 2022, to August 18, 2022. Please check the study period in the text and the tables.

Response: Thank you, we have corrected this typo.

R3.8. The description of the inclusion and exclusion criteria needs to be clarified and the exclusion flowchart may be improved. What was the recommended time interval between the first and the second booster? Please add this information to the inclusion criteria.

Response: Thank you for the suggestion. We have improved Extended Data Fig.1, have clarified inclusion and exclusion criteria, and have clarified that the recommended time interval between the first and second booster was 20 weeks.

R3.9. There are some discrepancies between the flowchart and its description in the main text. In the text, the authors indicated “study cohort included adults aged ≥ 20 years and affiliated with the Fondo Nacional de Salud (FONASA), the public national healthcare system, who completed a CoronaVac or BNT162b2’s two-dose primary series at least 120 days before the beginning of the follow-up on August 11, 2021, when the first-booster campaign was launched, and unvaccinated individuals “In the flowchart, the description is different, ...” at least 120 days before August 8, 2021” Please check the dates. Also, please check the number of vaccinated in the flowchart, it is different from the one previously reported in the text.

Response: Thank you for noting these discrepancies. We have updated Extended Data Figure 1 and checked that the reported numbers and dates are consistent with the main text and tables in the manuscript.

R3.10 In the footnote to the flowchart, the authors stand that “1,788 persons received three doses of CoronaVac plus a homologous CoronaVac second booster. We did not include the latter in the study due to the small sample size”. If individuals were excluded from all analyses, it should be mentioned in the exclusion criteria in the main text as well. Please clarify this point.

Response: Thank you for suggestion. We added this exclusion criterion in the main text and Extended Data Figure 1.

R3.11. The authors stand that they excluded individuals with probable or confirmed COVID-19 reported before February 15, 2022. How the “probable” COVID-19 case is defined? Please clarify this point and add the information on the number of excluded due to the previous infection to the flowchart - Extended Fig.1.

Response: Thank you for noting this inconsistency. We excluded individuals with confirmed COVID-19 according to reverse-transcription polymerase-chain-reaction assay for SARS-CoV-2 or antigen test. We have corrected this in the main text and the footnotes to Extended Data Figure 1.

R3.12. The description of variables included in the Cox model is not consistent First, the authors said: “Hazard ratios were adjusted for age, sex, region of residence, nationality, income, and comorbidities conditions (Extended Data Tables 1-5)”. Later the authors mentioned that “We show the adjusted vaccine effectiveness results, including covariates as controls (age, gender, region, nationality, health insurance category, and comorbidities)”. Please clarify if the income or the health insurance category or both were included in the model.

Response: Thank you for noting this inconsistency in our description of covariates for the Cox model. The error reflects an early version of the analysis when we did not have access to income data and used the health insurance category as a proxy for income. We used income in the analysis and have corrected the methods section to reflect this (p.16, lines 349-351).

R3.13 In the text the authors mention variable age (that is usually interpreted as age in years), in tables, the authors present the variable Age group. Please check this point.

Response: Thank you; we realize this was unclear. The analysis uses individual’s age as a continuous variable. However, we show age groups in the tables with descriptive statistics so that

readers get a better sense of the age distribution in the final sample. We have now clarified this in a footnote in the tables with descriptive statistics.

R3.14 Also, the authors named the region of residence as one of the covariates under study. Later, the authors renamed this covariate as a Cohort location. Please standardize the nomenclature in the text and the tables.

Response: Thank you; we have corrected this inconsistency.

R3.15. In the statistical analysis description, it is not clear which variables were used as a strata in the Cox models and how Cox model assumptions were checked. Please add this information to the online methods sections.

Response: Thank you for the suggestion; we agree this is important. We fit a stratified version of the extended Cox proportional-hazards model, stratifying by age, sex, region of residence, income, nationality, and whether the patient had underlying conditions that have been associated with severe Covid-19. Under this model, each combination of the predictors has a specific hazard function that can independently evolve. The variables used as strata are described in the Extended Data Tables 1-5. To address the reviewer's comment, we have now added this clarification, and also we describe how Cox model assumptions were checked in the online methods section as follows:

p.15, lines 334-335: "The model was stratified by age, sex, region of residence, income, nationality, and whether the patient had underlying conditions that have been associated with severe Covid-19, based on strata shown in Extended Data Tables 1-5."

p.17, lines 356-361: "The stratified version of the extended Cox proportional-hazards model assumes: (i) that the units in each stratum have a different baseline hazard rate, which depends on covariates shown in Extended Data Tables 1-5 (i.e., age, sex, region of residence, income, nationality, and whether the patient had underlying conditions that have been associated with severe

Covid-19), (ii) the coefficients for all these covariates do not change with time, (iii) vaccination status changes with time, and (iv) vaccine effectiveness is constant throughout the period considered for each estimate.”

R3.16. To estimate VE, the authors used unvaccinated as a reference category. It would be also important to understand the additional protection conferred by the second booster dose compared to the first booster. Please consider reporting relative VE using the first booster as a reference category.

Response: Thank you for the suggestion. We have now included Extended Data Table 8, showing the overall weekly effectiveness against ICU admissions and death for adults >20 years of mRNA-based second vaccine boosters compared to three-dose background regimens in Chile, March 7, 2022, through August 15, 2022. We have also included a graph showing these data (Extended Data Fig. 4). These results suggest that a second vaccine booster effectively prevents against COVID-19-related ICU admission and death and provides additional protection compared to the first booster. We also added this clarification in p.6, lines 125-131. (please see the Extended Data table and figure in our response R2.2).

R3.17. The authors assessed change in VE over time, estimating VE on a weekly basis. However, the weekly number of COVID-19-related severe outcomes (ICU admissions and deaths) is low, in particular when specific regimens are considered, leading to a lack of precision. Confidence intervals may be quite wide (for instance VE=85%, CI95%:0.0-98.0), as the authors documented in the study limitations. The assessment of change in VE over time is important and without a doubt can be one of the study’s strengths. The authors should consider to change the time window, a month instead of a week will be more appropriate to estimate VE for severe outcomes and it will improve the estimates precision.

Response: Thank you for the suggestion and the encouraging comments. While a longer time window, such as a month, may provide a more comprehensive estimate for severe outcomes, as the study progresses the precision of the estimates improves, particularly towards the end of the study.

References included in our response to referees

- 1 Dong, E., Du, H. & Gardner, L. An interactive web-based dashboard to track COVID-19 in real time. *Lancet Infect Dis* **20**, P533-P534 (2020).
- 2 Madhi, S. A. *et al.* Population Immunity and Covid-19 Severity with Omicron Variant in South Africa. *N Engl J Med* (2022).
- 3 Andrews, N. *et al.* Covid-19 Vaccine Effectiveness against the Omicron (B.1.1.529) Variant. *N Engl J Med*, doi:10.1056/NEJMoa2119451 (2022).
- 4 Collie, S., Champion, J., Moultrie, H., Bekker, L.-G. & Gray, G. Effectiveness of BNT162b2 vaccine against omicron variant in South Africa. *N Engl J Med* **386** (2022).
- 5 Chemaitelly, H. *et al.* Waning of BNT162b2 Vaccine Protection against SARS-CoV-2 Infection in Qatar. *N Engl J Med*, doi:10.1056/NEJMoa2114114 (2021).
- 6 Levin, E. G. *et al.* Waning Immune Humoral Response to BNT162b2 Covid-19 Vaccine over 6 Months. *N Engl J Med* **385**, e84, doi:10.1056/NEJMoa2114583 (2021).
- 7 Higdon, M. M. *et al.* Duration of effectiveness of vaccination against COVID-19 caused by the omicron variant. *Lancet Infect Dis* **22**, 1114-1116 (2022).
- 8 Jara, A. *et al.* Effectiveness of homologous and heterologous booster doses for an inactivated SARS-CoV-2 vaccine: a large-scale prospective cohort study. *Lancet Glob Health* **10**, 798-806, doi:10.1016/S2214-109X(22)00112-7 (2022).
- 9 Bar-On, Y. M. *et al.* Protection of BNT162b2 Vaccine Booster against Covid-19 in Israel. *N Engl J Med* **385**, 1393-1400, doi:10.1056/NEJMoa2114255 (2021).
- 10 Arbel, R. *et al.* BNT162b2 Vaccine Booster and Mortality Due to Covid-19. *N Engl J Med*, online first, doi:10.1056/NEJMoa2115624 (2021).
- 11 Barda, N. *et al.* Effectiveness of a third dose of the BNT162b2 mRNA COVID-19 vaccine for preventing severe outcomes in Israel: an observational study. *Lancet* **398**, 2093-2100 (2021).
- 12 Patalon, T. *et al.* Waning effectiveness of the third dose of the BNT162b2 mRNA COVID-19 vaccine. *Nature Communications* **13**, 3203, doi:10.1038/s41467-022-30884-6 (2022).
- 13 Ferdinands, J. M. *et al.* Waning 2-Dose and 3-Dose Effectiveness of mRNA Vaccines Against COVID-19-Associated Emergency Department and Urgent Care Encounters and Hospitalizations Among Adults During Periods of Delta and Omicron Variant Predominance - VISION Network,

- 10 States, August 2021-January 2022. *MMWR Morb Mortal Wkly Rep* **71**, 255-263, doi:10.15585/mmwr.mm7107e2 (2022).
- 14 Grewal, R. *et al.* Effectiveness of a fourth dose of covid-19 mRNA vaccine against the omicron variant among long term care residents in Ontario, Canada: test negative design study. *BMJ* **378**, e071502 (2022).
- 15 Jara, A. *et al.* Effectiveness of an Inactivated SARS-CoV-2 Vaccine in Chile. *N Engl J Med* **385**, 875-884, doi:10.1056/NEJMoa2107715 (2021).
- 16 Link-Gelles, R. *et al.* Effectiveness of 2, 3, and 4 COVID-19 mRNA vaccine doses among immunocompetent adults during periods when SARS-CoV-2 Omicron BA. 1 and BA. 2/BA. 2.12. 1 sublineages predominated—VISION Network, 10 states, December 2021–June 2022. *Morb Mortal Wkly Rep* **71**, 931 (2022).
- 17 Moreira Jr, E. D. *et al.* Safety and efficacy of a third dose of BNT162b2 Covid-19 vaccine. *N Engl J Med* **386**, 1910-1921 (2022).
- 18 Baden, L. R. *et al.* Phase 3 trial of mRNA-1273 during the Delta-variant surge. *N Engl J Med* **385**, 2485-2487 (2021).

REVIEWER COMMENTS

Reviewer #2 (Remarks to the Author):

The authors have satisfactorily addressed all my concerns.

Reviewer #3 (Remarks to the Author):

The authors addressed most of the comments adequately.

1. Title: The authors improved the formulation, but an indication of the main outcomes (ICU admission and death) is still missing. Please specify in the title that COVID-19 VE is estimated against ICU admission and death

2. Main text: Second paragraph, lines 61-67 the authors' statement is now not up-to-date, currently there is a lot of evidence on the second boosters vaccine effectiveness as well as on a waning of protection against COVID-19 for second boosters. The evidence might exist and the authors might be not aware of it, so "no evidence" is a very strong statement.

Please reformulate the following paragraph, highlighting the idea of comparisons of different schedules and vaccines as well as weekly estimation:

However, there is limited evidence for the effectiveness of second boosters, and it primarily refers to mRNA vaccines in Israel, Canada, and the USA among populations at higher risk (e.g., older adults and immunocompromised persons). Furthermore, there is no evidence of a potential waning of protection against COVID-19 for second boosters, or the effectiveness against severe disease of an second booster dose for individuals who received their primary series based on inactivated vaccines, which constitute a considerable proportion of vaccinated individuals in low- and middle-income countries.

3. As suggested the authors added relative VE estimates (reference group vaccinated with first booster), this also needs to be reflected in the study methods. It is not clear whether the authors defined this reference group (all individuals with the first booster regardless of the time of the booster uptake or only those with at least 20 weeks after the first booster uptake were included in this group). Please clarify.

4. Line 121 Please start a new paragraph when describing VE estimates.

5. Please add references to support the following statement in lines 200-203 ...The introduction of the BA.4 and BA.5 Omicron sub-lineages during the second part of the study

period and immunity waning due to a decline in the circulating levels of neutralizing antibodies as described for primary series and first booster doses, may explain the decrease in protection.

6. In Table 7. For the week of 27 of June VE of 100% is reported but the respective confidence interval is inconsistent with the point estimate (77.3-96.3) Please check all the estimates.

7. In line 108 of the main text the authors stated that ... Our final cohort included 3,754,785 adults. Of these, 2,623,802 (69.9) received a second booster shot. In the exclusion diagram, the authors split the overall number of vaccinated with 2nd booster by vaccination scheme and have the following figures:

$237062+841122+137165+1394424+1788= 2611561$ instead of 2326802, 12241 participants are missing

Please check again the numbers in the exclusion diagram and in the text

8. Please standardize "COVID-19 related" and "COVID-19-related" along with the manuscript

9. The authors estimated VE weekly, it is not clear from the manuscript why so short time window was used. It is also not clear if the weekly VE monitoring was prospective, i.e. every week VE was estimated to inform health authorities or the authors retrospectively estimated VE by week. If the authors describe a prospective monitoring system it should be clearly highlighted. If the aim was to estimate wain over time, estimation by time since the second booster will be a more informative approach, and will allow formally test the statistical significance of the differences in VE over time.

In addition to weekly VE estimates that are currently reported, please provide in supplementary material the number of events, person-days and crude hazard ratio with confidence interval for each weekly estimate.

Response to referees' comments

Nature Communications NCOMMS-22-40863A

Effectiveness and duration of a second COVID-19 vaccine booster: a large-scale prospective cohort study during the SARS-CoV-2 Omicron outbreak in Chile

Date: July 04, 2023

We are grateful for the additional thoughtful comments and suggestions from reviewer 3. We think the quality of our manuscript has improved as a result of this revision. The referees' comments are numbered consecutively, reproduced verbatim, and our responses are labeled as "Response" and highlighted in yellow. Additional references are listed at the end of this document. We have made minor editorial and formatting updates to enhance the readability and overall presentation of the manuscript.

Reviewer #1

No additional comments.

Reviewer #2

R2.1 The authors have satisfactorily addressed all my concerns.

Response: Thank you.

Reviewer #3

R31. The authors addressed most of the comments adequately.

Response: Thank you.

R3.2. Title: The authors improved the formulation, but an indication of the main outcomes (ICU admission and death) is still missing. Please specify in the title that COVID-19 VE is estimated against ICU admission and death

Response: Thank you for the suggestion. Nature Communications recommends that titles have 15 words or fewer and should not contain abbreviations. Our article went from nine to 21 words following your suggestion in the first round of comments to include time, place, and study design. Adding, for example, "...against intensive care unit admission and death" would leave our title 28 words long, almost double the length recommended by the editor. We would be happy to modify the title, if the editor deems it necessary, to "Effectiveness and duration of a second vaccine booster against COVID-19-related intensive care unit admission and death: a large-scale prospective cohort

study during the SARS-CoV-2 Omicron outbreak in Chile”. To partially address the reviewer’s concern, we have added the main outcomes as keywords.

R3.3. Main text: Second paragraph, lines 61-67 the authors’ statement is now not up-to-date, currently there is a lot of evidence on the second boosters vaccine effectiveness as well as on a waning of protection against COVID-19 for second boosters. The evidence might exist and the authors might be not aware of it, so “no evidence “ is a very strong statement. Please reformulate the following paragraph, highlighting the idea of comparisons of different schedules and vaccines as well as weekly estimation: However, there is limited evidence for the effectiveness of second boosters, and it primarily refers to mRNA vaccines in Israel, Canada, and the USA among populations at higher risk (e.g., older adults and immunocompromised persons). Furthermore, there is no evidence of a potential waning of protection against COVID-19 for second boosters, or the effectiveness against severe disease of an second booster dose for individuals who received their primary series based on inactivated vaccines, which constitute a considerable proportion of vaccinated individuals in low- and middle-income countries.

Response: Thank you for your suggestions. We have updated our references related to second boosters and waning protection of Covid-19 vaccines, including studies from Canada, Israel, Korea, Portugal, Sweden, and the USA among individuals with mRNA vaccine boosters, and Thailand among individuals with heterologous three-dose vaccine regimes, including inactivated virus vaccines.¹⁻¹¹ We toned down the statement about “no evidence” and reformulated the paragraph following the reviewer’s suggestion. Last, we expanded the discussion to include more recent references. Please see pp.3-4, lines 63-71, pp. 8-9, lines 172-187.

R3.4. As suggested the authors added relative VE estimates (reference group vaccinated with first booster), this also needs to be reflected in the study methods. It is not clear whether the authors defined this reference group (all individuals with the first booster regardless of the time of the booster uptake or only those with at least 20 weeks after the first booster uptake were included in this group). Please clarify.

Response: Thank you for the suggestion. We have now clarified this in the main text (p.4, lines 81-82), and p.17, lines 391-394, “To understand additional protection conferred by the second booster compared to the first booster, we also considered individuals with a three-dose regime as a comparison group. In this case, we considered the period 13 days after the first booster dose, regardless of the time of the booster uptake”, and minor clarifications in the rest of the methods section (e.g., p.18, lines 397-399).

R3.5 Line 121 Please start a new paragraph when describing VE estimates.

Response: Done.

R3.6 Please add references to support the following statement in lines 200-203 ...The introduction of the BA.4 and BA.5 Omicron sub-lineages during the second part of the study period and immunity waning

due to a decline in the circulating levels of neutralizing antibodies as described for primary series and first booster doses, may explain the decrease in protection.

Response: Thank you for the suggestion. The section now reads: “The introduction of the BA.4 and BA.5 Omicron sub-lineages during the second part of the study period (Extended Data Figure 3) and immunity waning due to a decline in the circulating levels of neutralizing antibodies as described for primary series and first booster doses, may explain the decrease in protection.”¹²⁻¹⁴

R3.7. In Table 7. For the week of 27 of June VE of 100% is reported, but the respective confidence interval is inconsistent with the point estimate (77.3-96.3) Please check all the estimates.

Response: Thank you for bringing this error to our attention. We have corrected the typo and checked all other estimates.

R3.8. In line 108 of the main text the authors stated that ... Our final cohort included 3,754,785 adults. Of these, 2,623,802 (69.9) received a second booster shot. In the exclusion diagram, the authors split the overall number of vaccinated with 2nd booster by vaccination scheme and have the following figures: 237,062+841,122+137,165+1,394,424+1,788= 2,611,561 instead of 2,326,802, 12,241 participants are missing

Please check again the numbers in the exclusion diagram and in the text.

Response: Thank you for bringing this oversight to our attention. In the article, we evaluated the effectiveness of second vaccine boosters in preventing severe illness for four different three-dose background regimes: BNT162b2 primary series plus a homologous booster, and CoronaVac primary series plus an mRNA booster, a homologous booster, and a ChAdOx-1 booster. BNT162b2 and Coronavac were the two most important primary series in Chile. The 12,241 missing participants with a second booster were excluded because they had received a primary series of Oxford-AstraZeneca’s ChAdOx1 adenoviral vector vaccine or CanSinoBIO’s Ad5-nCoV. We have now clarified this in the manuscript:

(p.5, lines 111-113) “We also excluded individuals with a second vaccine booster who had received a primary series of Oxford-AstraZeneca’s ChAdOx1 adenoviral vector vaccine or CanSinoBIO’s Ad5-nCoV (n=12,241).”

Extended Data Fig.1. Study participants and cohort eligibility. Participants were adults aged ≥ 20 years affiliated with the Fondo Nacional de Salud (FONASA), the public national healthcare system in Chile, who completed a Coronavac or BNT162b2’s two-dose primary series at least 120 days before the beginning of the follow-up on August 11, 2021, when the first-booster campaign was launched and unvaccinated individuals. We excluded individuals with confirmed COVID-19 according to reverse-transcription polymerase-chain-reaction assay for SARS-CoV-2 or antigen test reported before August 11, 2021. We excluded individuals who received three doses of CoronaVac plus a homologous CoronaVac second booster due to the small sample size ($n=1,788$) and individuals with a second vaccine booster who had received a primary series of ChAdOx1 or Ad5-nCoV ($n=12,241$).

R3.9. Please standardize “COVID-19 related” and “COVID-19-related” along with the manuscript.

Response: Corrected, thank you.

R3.10 The authors estimated VE weekly, it is not clear from the manuscript why so short time window was used. It is also not clear if the weekly VE monitoring was prospective, i.e. every week VE was estimated to inform health authorities or the authors retrospectively estimated VE by week. If the authors describe a prospective monitoring system it should be clearly highlighted. If the aim was to estimate wain

over time, estimation by time since the second booster will be a more informative approach, and will allow formally test the statistical significance of the differences in VE over time.

In addition to weekly VE estimates that are currently reported, please provide in supplementary material the number of events, person-days, and crude hazard ratio with confidence interval for each weekly estimate.

Response: As you suggest, this article reflects our work as collaborators or public health officials from the Ministry of Health. We gave priority to requests from the Minister of Health herself, aiming to make urgent policy decisions during the pandemic in Chile. In early 2022, several countries initiated the administration of second boosters (a fourth dose for most vaccines) at intervals of four to six months after the first booster dose. International organizations recommended this approach for at-risk populations.²²⁻²⁶ Our objective was to estimate the effectiveness of second vaccine boosters in preventing severe illness for four different three-dose background regimes, considering the most prevalent primary series in Chile: BNT162b2 and Coronavac. We provided weekly reports on the vaccine effectiveness of booster shots to the Ministry of Health through a prospective monitoring system as you suggest. However, our point estimates are not weekly estimates. They show the estimated vaccine effectiveness throughout the period considered, as described in the methods. We appreciate your suggestion regarding the inclusion of the time elapsed since the last vaccine dose when assessing the waning of immunity. It is indeed an important aspect to consider. However, we had specific reasons for not incorporating this element in our current study. Our primary objective was to generate summary statistics of vaccine effectiveness that policymakers could use to track the overall protection levels across different populations and stages of the vaccination campaign. We aimed to provide a comprehensive perspective on vaccine efficacy, encompassing various vaccination schedules and vaccine types, with a particular focus on vaccine schedules commonly used in low- and middle-income countries. While an analysis based on the time since the last vaccine dose could provide valuable insights into the potential decay of immunity over time, it would also introduce complexity and limit the comparability of our findings across different populations and timeframes. Vaccination campaigns are inherently dynamic, with varying intervals between vaccine doses across regions. This variability makes it challenging to establish a consistent timeframe to evaluate the decline of immunity. To address the concern raised by the reviewer, we have now clarified these points in the manuscript in p.9, lines 197-204).

Thank you for time and suggestions and suggestions to improve our manuscript.

References included in our response to referees

- 1 World Health Organization. Good practice statement on the use of second booster doses for COVID 19 vaccines. 12 (World Health Organization, Geneva, 2022).
- 2 Grewal, R. *et al.* Effectiveness of a fourth dose of covid-19 mRNA vaccine against the omicron variant among long term care residents in Ontario, Canada: test negative design study. *BMJ* **378**, e071502 (2022).

- 3 Link-Gelles, R. *et al.* Effectiveness of 2, 3, and 4 COVID-19 mRNA vaccine doses among immunocompetent adults during periods when SARS-CoV-2 Omicron BA. 1 and BA. 2/BA. 2.12. 1 sublineages predominated—VISION Network, 10 states, December 2021–June 2022. *Morbidity and Mortality Weekly Report* **71**, 931 (2022).
- 4 Nordström, P., Ballin, M. & Nordström, A. Effectiveness of a fourth dose of mRNA COVID-19 vaccine against all-cause mortality in long-term care facility residents and in the oldest old: A nationwide, retrospective cohort study in Sweden. *The Lancet Regional Health–Europe* **21** (2022).
- 5 Magen, O. *et al.* Fourth dose of BNT162b2 mRNA Covid-19 vaccine in a nationwide setting. *New England Journal of Medicine* **386**, 1603-1614 (2022).
- 6 Arbel, R. *et al.* Effectiveness of a second BNT162b2 booster vaccine against hospitalization and death from COVID-19 in adults aged over 60 years. *Nature medicine* **28**, 1486-1490 (2022).
- 7 Gazit, S. *et al.* Short term, relative effectiveness of four doses versus three doses of BNT162b2 vaccine in people aged 60 years and older in Israel: retrospective, test negative, case-control study. *Bmj* **377** (2022).
- 8 Park, S. K. *et al.* Effectiveness of A Fourth Dose of COVID-19 mRNA Vaccine in the Elderly Population During the Omicron BA.2 and BA.5 Circulation: A Nationwide Cohort Study in Korea (K-COVE). *Open Forum Infectious Diseases* **10**, doi:10.1093/ofid/ofad109 (2023).
- 9 Kislaya, I. *et al.* COVID-19 mRNA vaccine effectiveness (second and first booster dose) against hospitalisation and death during Omicron BA. 5 circulation: cohort study based on electronic health records, Portugal, May to July 2022. *Eurosurveillance* **27**, 2200697 (2022).
- 10 Chariyalertsak, S. *et al.* Effectiveness of heterologous 3rd and 4th dose COVID-19 vaccine schedules for SARS-CoV-2 infection during delta and omicron predominance in Thailand. *Research Square*, Preprint 28 jun, doi:10.21203/rs.3.rs-1792139/v1 (2022).
- 11 Intawong, K. *et al.* Reduction in severity and mortality in COVID-19 patients owing to heterologous third and fourth-dose vaccines during the periods of delta and omicron predominance in Thailand. *International Journal of Infectious Diseases* **126**, 31-38, doi:<https://doi.org/10.1016/j.ijid.2022.11.006> (2023).
- 12 Cele, S. *et al.* Omicron extensively but incompletely escapes Pfizer BNT162b2 neutralization. *Nature* **602**, 654-656, doi:10.1038/s41586-021-04387-1 (2022).
- 13 Garcia-Beltran, W. F. *et al.* mRNA-based COVID-19 vaccine boosters induce neutralizing immunity against SARS-CoV-2 Omicron variant. *Cell* **185**, 457-466. e454 (2022).
- 14 Barouch, D. H. Covid-19 Vaccines — Immunity, Variants, Boosters. *New England Journal of Medicine* **387**, 1011-1020, doi:10.1056/NEJMra2206573 (2022).